# Anti-Inflammatory Effects of *Spirodela polyrhiza* (L.) S_CHLEID_. Extract on Contact Dermatitis in Mice—Its Active Compounds and Molecular Targets

**DOI:** 10.3390/ijms241713271

**Published:** 2023-08-26

**Authors:** Kukhwa Kim, Daniel Lee, Han-Young Kim, Soyeon Kim, Ji-Hyo Lyu, Sujung Park, Young-Chul Park, Hyungwoo Kim

**Affiliations:** 1Department of Sasang Constitutional Medicine, Pusan National University Korean Medicine Hospital, Yangsan 50612, Republic of Korea; kimkh1243@gmail.com; 2Division of Pharmacology, School of Korean Medicine, Pusan National University, Yangsan 50612, Republic of Korea; skekeksl@naver.com (D.L.); whatsnikers00@pusan.ac.kr (H.-Y.K.); 3Research Institute for Korean Medicine, Pusan National University, Yangsan 50612, Republic of Korea; amugdale@pusan.ac.kr (S.K.); sujupark@pusan.ac.kr (S.P.); 4Herbal Medicine Resources Research Center, Korea Institute of Oriental Medicine, Naju 58245, Republic of Korea; jhlyu@pusan.ac.kr; 5Department of Microbiology & Immunology, School of Medicine, Pusan National University, Yangsan 50612, Republic of Korea; ycpark@pusan.ac.kr

**Keywords:** *Spirodela polyrhiza*, traditional medicine, herbal medicine, inflammation, skin, network pharmacology

## Abstract

*Spirodela polyrhiza* (L.) S_CHLEID_. has been used to treat epidemic fever, dysuria, and various skin ailments, such as measles eruptions, eczema, and pruritus, in China, Japan, and Korea. In this study, the active compounds in *S. polyrhiza* and their target genes were identified by network-based analysis. Moreover, the study evaluated the effects of a 70% ethanolic extract of *S. polyrhiza* (EESP) on skin lesions, histopathological changes, inflammatory cytokines, and chemokines in mice with contact dermatitis (CD) induced by 1-fluoro-2,4-dinitrobenzene (DNFB), and examined the inhibitory effects of EESP on mitogen-activated protein kinase (MAPK) signalling pathways. In our results, 14 active compounds and 29 CD-related target genes were identified. Among them, tumour necrosis factor (TNF) and interleukin 6 (IL-6) were identified as hub genes, and luteolin and apigenin showed a strong binding affinity with TNF (<−8 kcal/mol) and IL-6 (<−6 kcal/mol). Our in vivo studies showed that topical EESP ameliorated DNFB-induced skin lesions and histopathological abnormalities, and reduced the levels of TNF-α, interferon (IFN)-ɣ, IL-6, and monocyte chemotactic protein (MCP)-1 in inflamed tissues. In conclusion, our findings suggest the potential for dermatological applications of *S. polyrhiza* and suggest that its anti-dermatitis action is related to the inhibition of TNF and IL-6 by luteolin and luteolin glycosides.

## 1. Introduction

Contact dermatitis (CD) has a long history, dating back to ancient Egypt, as evidenced by its appearance on papyrus scrolls. Documentation of CD has been found in many different cultures and countries, not only in Europe but also in China and India [1]. At present, CD can have a significant impact on quality of life, and 20.1% of the general population have contact allergies to certain substances [2]. Occupational skin diseases (OSD) contribute to around 25% of all work-related lost days, and CD is responsible for 70–90% of OSD cases [3,4]. CD causes itching and blistered, dry, cracked skin, and can cause lighter skin to redden and darker skin to become dark brown, purple, or grey [5].

Corticosteroids are considered the gold-standard for managing acute CD flare-ups and, when used appropriately, are generally safe for short-term use. However, the long-term use or misuse of potent corticosteroids can lead to adverse effects such as skin thinning, discoloration, and increased infection risk. In instances involving unavoidable and repetitive use, such as OCD, this can pose a greater challenge. Thus, novel therapeutic agents are required, and complementary and alternative medicines based on natural substances offer relatively safe alternatives [6,7].

*Spirodela polyrhiza* (L.) S_CHLEID_., commonly known as duckweed, is a species of aquatic flowering plant belonging to the family Araceae. Duckweeds are small, free-floating plants found in ponds, lakes, slow-moving streams, and other freshwater bodies worldwide. In East Asian literature, duckweed is often used as a metaphor for a fleeting or wandering life. Duckweeds, including this species, have been well-studied because of their ability to remove pollutants from wastewater, which makes them a promising tool for water treatment. Additionally, they can be used as a food source for livestock, poultry, and fish due to their high protein contents [8].

In traditional East Asian medicine, whole *S. polyrhiza* is a diaphoretic with a cool property and is used to induce sweating, reduce fever, and as a diuretic and detoxifier. Traditional medicine experts in China, Japan, and Korea have used this plant to treat epidemic fever, dysuria, and skin ailments, such as measles eruption, eczema, and pruritus [9]. Furthermore, *S. polyrhiza* has been reported to have anti-inflammatory [10] and anti-allergic effects [11,12,13].

We aimed to investigate whether the candidate plant found in traditional Korean medicine has therapeutic effects on CD and to explore its active compounds and associated mechanisms. In this study, active compounds of *S. polyrhiza* and their targets were identified using network pharmacological analysis. In addition, the anti-inflammatory efficacy of a 70% ethanolic extract of whole *S. polyrhiza* (EESP) was investigated in a murine model of CD.

## 2. Result

### 2.1. Fourteen Active Compounds Were Found in S. polyrhiza

In total, 73 compounds were found in the TCMSP, ETCM and TCMBank databases (Appendix A), and target gene list of 730 was obtained (Appendix A). Of these, 23 compounds had CD-related gene targets as determined by the DisGeNET database and 14 compounds ((+)-armepavine, (R)-*N*-methylcoclaurine, 2-hexadecenoic acid, anonaine, apigenin, butylated hydroxytoluene, D-ascorbic acid, higenamine, isoliensinine, luteolin, nornuciferine, nuciferine, palmitic acid and roemerine) satisfied the ADME criteria. Among them, six compounds, including apigenin, D-ascorbic acid and luteolin, had a DrugBank (https://go.drugbank.com/ (accessed on 16 August 2023)) accession number (Table 1).

### 2.2. Twenty-Nine CD-Related Targets Were Found in S. polyrhiza

To identify CD-related targets of *S. polyrhiza*, a list of disease-associated genes was extracted from the DisGeNET platform. A total of 110 genes were found, of which 107 genes were identified (evidence index > 0.7). Using the UniProt database, the names of the targets of all compounds (730 targets) were checked. A comparison with 107 CD-related genes resulted in the identification of 29 CD-related target genes (aryl hydrocarbon receptor (AHR), BCL2 apoptosis regulator (BCL2), caspase 8 (CASP8), C-C motif chemokine ligand 2 (CCL2), cellular communication network factor 2 (CCN2), cytochrome P450 family 1 subfamily A member 1 (CYP1A1), DNA damage-inducible transcript 3 (DDIT3), EPH receptor B2 (EPHB2), glutamate–cysteine ligase catalytic subunit (GCLC), glutathione-disulfide reductase (GSR), glutathione S-transferase pi 1 (GSTP1), heme oxygenase 1 (HMOX1), interferon gamma (IFNG), interleukin 1 alpha (IL1A), interleukin 1 beta (IL1B), interleukin 4 (IL4), interleukin 6 (IL6), mitogen-activated protein kinase 1 (MAPK1), matrix metallopeptidase 10 (MMP10), opioid receptor mu 1 (OPRM1), S100 calcium binding protein A8 (S100A8), superoxide dismutase 1 (SOD1), superoxide dismutase 3 (SOD3), toll like receptor 4 (TLR4), tumor necrosis factor (TNF), tryptase alpha/beta 1 (TPSAB1), UDP glucuronosyltransferase family 1 member A1 (UGT1A1), UDP glucuronosyltransferase family 1 member A6 (UGT1A6), vascular endothelial growth factor A (VEGFA)) (Appendix A). Correlations between the 29 target genes were confirmed using the STRING platform. The results showed that 25 genes (nodes) had 65 edges with an interaction score greater than 0.7 (high confidence) and 4 genes had no edges (Figure 1). When four different methods were applied using the CytoHubba plugin, the results ranked IL6 and TNF as the top candidates, followed by IL1B and IL4 (Appendix A).

### 2.3. Fourteen Active Compounds and Twenty-One CD-Related Targets Were Closely Associated

The results of identifying target genes for 14 active compounds revealed that, among the 29 CD-related genes shown in Figure 1, 21 genes were identified. Correlations between 14 active compounds and 21 genes were analysed using Cytoscape 3.91. The results showed that apigenin interacted with 11 disease targets, while D-ascorbic acid (vitamin C) and luteolin were associated with 10 and 8, respectively. In addition, TNF was associated with five active compounds, including apigenin, D-ascorbic acid, and luteolin. IL6 was connected to D-ascorbic acid and luteolin. IFNG was correlated with apigenin and luteolin. Lastly, monocyte chemotactic protein-1 (MCP-1, (CCL2)) was found to be associated with D-ascorbic acid (Figure 2).

### 2.4. Eleven and Eight Active Compounds Showed Strong Binding Affinity to TNF or IL-6, Rrespectively

Binding affinities between TNF or IL-6 and the 14 active compounds were predicted by molecular docking analysis (Table 2). Eleven compounds showed strong binding affinity with TNF with binding energies of <−7 kcal/mol, and eight compounds showed strong binding affinity with IL-6 with binding energies of <−6 kcal/mol. The binding modes of the active compounds and residues of TNF or IL-6 are shown in Figure 3A,B.

### 2.5. Apigenin and Luteolin Were Identified in EESP

High-performance liquid chromatography (HPLC) detected four peaks in EESP, which were attributed to apigenin, apigenin-7-glucoside, luteolin, and luteolin-7-glucoside using commercial standards. The retention times of the four standard materials were 21.886 (luteolin-7-glucoside), 27.570 (apigenin-7-glucoside), 36.369 (luteolin), and 38.530 min (apigenin), respectively (Figure 4).

### 2.6. EESP Ameliorated Skin Lesions and Inhibited Skin Thickening in CD Mice

The repeated topical application of 1-fluoro-2,4-dinitrobenzene (DNFB) resulted in the development of skin lesions characteristic of CD that exhibited skin scaling, excoriation, erythema, and roughness (Figure 5A,B). Topical EESP administered for 6 days provided relief from these symptoms, as depicted in Figure 5A. Additionally, when treated with EESP (150 or 500 μg/day), skin surface scores were significantly reduced (Figure 5B); notably, mean skin weight was considerably greater in the CTL group than in the NOR group, whereas the application of EESP effectively prevented skin weight increases (Figure 5C). The application of DNFB resulted in a significant increase in melanin and erythema indices in the CTL group; when EESP was applied at 500 μg/day, there was a notable reduction in melanin indices, as shown in Figure 5D. Furthermore, EESP application at 150 or 500 μg/day significantly reduced erythema indices (Figure 5E).

### 2.7. EESP Prevented DNFB-Induced Histopathological Abnormalities

The repeated application of DNFB resulted in notable epidermal hyperplasia, hyperkeratosis, epidermis destruction, and immune cell infiltration (Figure 6A). Treatment with EESP at 150 or 500 μg/day significantly reduced epidermal hyperplasia (Figure 6B). Furthermore, EESP administration at 150 and 500 μg/day considerably reduced infiltrating immune cell numbers around dermis and blood vessels (Figure 6C and Appendix A).

### 2.8. EESP Effectively Reduced DNFB-Induced Increases in TNF-α, IFN-γ, IL-6, and MCP-1 Levels in Inflamed Tissues

TNF-α, IFN-γ, IL-6, and MCP-1 levels were significantly elevated in the DNFB-treated skins of mice in the CTL group. When EESP was topically applied at a dosage of 500 μg/day, it effectively reduced these DNFB-induced increases in inflammatory cytokine and chemokine levels. EESP at all concentrations administered significantly lowered TNF-α levels and, at 150 and 500 μg/day, EESP also significantly lowered IL-6 and MCP-1 levels (Figure 7).

### 2.9. EESP Suppressed the DNFB-Induced Phosphorylation of ERK

Repeated DNFB applications induced the phosphorylation of c-Jun N-terminal kinase (JNK), extracellular-signal-regulated kinase (ERK), and p38 MAPK in inflamed tissues. Topical EESP suppressed the DNFB-induced phosphorylation of ERK (Figure 8).

## 3. Discussion

In the network pharmacologic analysis, we identified 14 active compounds and 29 target genes in *S. polyrhiza*. Among them, IL6 and TNF were identified as hub genes. Among the 14 active compounds, apigenin and luteolin showed a strong binding affinity with both TNF and IL-6. Our in vivo studies showed that EESP ameliorated skin lesions and histopathological abnormalities in CD mice. In addition, EESP effectively reduced DNFB-induced increases in TNF-α, IFN-γ, IL-6, and MCP-1 levels in inflamed tissues. These results imply that *S. polyrhiza* can exert an anti-inflammatory effect in CD and that the therapeutic mechanism is related to the inhibition of cytokines such as TNF and IL-6.

The anti-inflammatory effects of *S. polyrhiza* have been established in several previous studies [10,13]. The fact that it effectively reduces the increase in TNF-α and IL-6 induced by lipopolysaccharide (LPS) stimulation in RAW264.7 cells [10], along with the significant decrease in TNF-α and IL-6 observed in an atopic dermatitis animal model [13], consistently aligns with the findings of our study.

Using a network pharmacologic approach, we identified 14 active compounds, including apigenin, D-ascorbic acid, and luteolin (Table 1), as well as 29 target genes, which included TNF, IL6, IL1B and IL4 in *S. polyrhiza*. These may serve as markers of the anti-inflammatory effect of *S. polyrhiza* on the skin. An analysis of correlations between the 29 target genes using the STRING platform and the cytoHubba plugin revealed that IL6 and TNF were identified as hub genes (Figure 1 and Appendix A). This indicates that TNF and IL6 should be given priority consideration for research on the anti-inflammatory effects of *S. polyrhiza*, owing to their high centrality.

For the identification of enriched transcriptomic signatures, analysis was conducted by using Enrichr website. The enrichment analysis of RNAseq automatic Gene Expression Omnibus (GEO) signatures revealed that active compounds of *S. polyrhiza* (730 target genes) were associated with mouse down gene set including Atopic Dermatitis Nestin-Cre IKK2 (GSE109936) [14] (Appendix A).

Among the 14 active compounds, apigenin, D-ascorbic acid and luteolin were directly linked to TNF (Figure 2). Among active compounds, apigenin, luteolin and butylated hydroxytoluene were linked to both TNF and HMOX1, eight compounds, except for the six compounds related to OPRM1, were linked to antioxidant or anti-inflammatory factors. These results mean that they can have antioxidant and anti-inflammatory effects, and that antioxidant and anti-inflammatory actions will play an important role in the therapeutic mechanism of *S. polyrhiza*.

Interestingly, OPRM1 and six active compounds did not correlate with other target genes or compounds (Figure 2). In addition, there was no connection between OPRM1 and other target genes in the protein–protein interaction (PPI) analysis (Figure 1). These results suggest the possibility of dividing the compound groups of *S. polyrhiza* related to the treatment of CD into two clusters: the itching improvement group (OPRM1-related 6 compounds) [15] and the antioxidant and anti-inflammatory group.

As shown in Table 2 and Figure 3, apigenin and luteolin could stably bind to TNF and IL-6. As apigenin is directly linked to TNF and luteolin is directly linked to both TNF and IL-6 in Figure 2, this result can be interpreted as the possibility that these two compounds bind to TNF or IL-6 and inhibit its function. Further research in this area will be needed. On the other hand, D-ascorbic acid had a binding affinity greater than -6 (Table 2) despite being directly linked to TNF and IL-6 (Figure 2). This seems to be because D-ascorbic acid indirectly inhibits TNF or IL-6 through its antioxidant effect. 

As shown in Figure 4, apigenin, apigenin-7-glucoside, luteolin and luteolin-7-glucoside were detected in EESP by HPLC, and the two peaks appearing before these four main peaks were assigned to luteolin-8-glucoside and apigenin-8-glucoside, respectively [16]. Apigenin and luteolin are standard substances and commonly used active compounds for the authentication of *S. polyrhiza* [11,17]. The *S. polyrhiza* extract used in this study contained a relatively large amount of luteolin compared to apigenin (Figure 4). This result implies that luteolin was mainly involved in the anti-inflammatory activity of the *S. polyrhiza* extract used in this experiment. 

In the present study, we found that the repeated application of EESP significantly improved scaling, erythema, and pigmentation, and reduced skin weights in our DNFB-induced mouse model (Figure 5). In addition, EESP also inhibited epidermal hyperplasia, immune cell infiltration, and hyperkeratosis (Figure 6). These observations suggest that topical EESP can suppress excessive epidermal cell growth, inhibit skin-thickening, and reduce abnormal keratinization. In addition, the skin-lightening effect that was observed raises the possibility that EESP has potential use as a skin-whitening agent and well-known inhibitors of melanogenesis, such as luteolin [18], may be involved in this action of EESP.

TNF-α and IFN-γ play crucial roles in the initiation and amplification of inflammatory responses, and both can activate keratinocytes, promote the release of pro-inflammatory mediators and disrupt skin barrier function [19], which causes hyperplasia, erosion, and hyperkeratosis in the epidermis [20]. IL-6 induces the production of other pro-inflammatory cytokines, such as IL-1β and TNF-α, by keratinocytes and modulates the production and distribution of key proteins involved in skin barrier integrity, such as filaggrin and tight junction proteins [21]. MCP-1 (also known as CCL2) functions as a chemoattractant for various immune cells, including monocytes, T cells, neutrophils, and dendritic cells [22]. In addition, IL-6, TNF-α, and IFN-γ act in concert to promote immune cell recruitment and inflammation by enhancing the expression of adhesion molecules in keratinocytes [19,20,21] and accelerating immune cell infiltration into inflamed tissues, thus amplifying the inflammatory response.

In the present study, EESP effectively lowered TNF-α, IFN-ɣ, IL-6, and MCP-1 levels in the inflamed skin tissues of CD mice (Figure 7) and inhibited skin-thickening and epidermal hyperplasia (Figure 5 and Figure 6). Our results indicate EESP suppresses skin-thickening by inhibiting pro-inflammatory cytokine and chemokine production, preventing keratinocyte accumulation, and thus suppressing epidermal hyperplasia. In addition, EESP significantly suppressed immune cell infiltration (Figure 6C), which we presume was due to its inhibition of pro-inflammatory cytokine and chemokine production. Collectively, these findings suggest that EESP can suppress the production of TNF-α, IFN-γ, IL-6, and MCP-1 and consequently inhibit epidermal hyperplasia, hyperkeratosis, and immune cell infiltration, which ameliorate the skin symptoms of CD and suppress skin-thickening. Considering the results in Figure 2, which show a connection between apigenin, D-ascorbic acid, and luteolin with TNF, IFNG, IL6, and CCL2 (MCP-1), it seems that these three main active compounds are associated with inhibitory effects on the production of cytokines and chemokine.

MAPK signalling plays an important role in the pathogenesis and progression of CD and cytokine secretion, including IL-6 and TNF-α [23,24]. Our results show that EESP prevented the DNFB-induced phosphorylation of ERK (Figure 8). In addition, Lee et al. reported that ethanol extracts of *S. polyrhiza* inhibited the JNK, ERK and p38 MAPK pathways in an animal model of atopic dermatitis [13]. These results imply that the anti-inflammatory mechanism initiated by EESP is involved in the regulation of cytokine production through inhibition of the MAPK pathway.

Luteolin can attenuate adverse photobiological effects on the skin and has anti-oxidative and anti-inflammatory effects on keratinocytes, fibroblasts, and several types of immune cells. In addition, luteolin can suppress pro-inflammatory mediators and regulate various signalling pathways [25]. We found that luteolin was correlated with 8 of the 29 selected disease target genes, including TNF, IFNG, and IL6 (Figure 2). In addition, luteolin showed strong binding affinities with both TNF and IL-6 (Table 2 and Figure 3). These findings indicate that luteolin underpins the anti-dermatitis effect of EESP and offers a starting point for the development of topical and systemic agents for inflammatory skin diseases.

We found that topical DEX administration suppressed weight gain and decreased spleen/body weight ratios, which is considered an indicator of general immune suppression [26], in CD mice, while EESP had no effect (Appendix A). These findings imply that the anti-inflammatory mechanism of EESP differs from that of corticosteroids, especially in terms of systemic immune suppression.

As shown in Figure 5D,E, DEX did not exhibit any improvement in skin color changes induced by DNFB. This phenomenon is often observed in experimental models of CD and commonly coincides with skin atrophy. Furthermore, there are instances where DEX fails to ameliorate surface symptoms on the skin [26,27,28]. The therapeutic mechanism of corticosteroids in eczema primarily involves their anti-inflammatory effects, which are achieved through immunosuppression rather than direct enhancement of the skin barrier function. On the other hand, it is worth noting that corticosteroid usage can lead to skin atrophy, potentially resulting in increased redness or darkening of the skin due to thinning. For this reason, while corticosteroids effectively suppress inflammation, the development of complementary agents aiming to restore skin integrity and improve the skin barrier function appears to be essential.

We did not perform pharmacokinetic studies. Therefore, there is no basis to determine whether the effect of EESP is local or systemic. Since EESP is applied to the skin, it is necessary to study how certain compounds are absorbed through the skin, how they are distributed, and how they are metabolized. In addition, because we have not checked all the compounds in EESP, it is hard to know what compounds are in the sample, how much of the sample they are present in, and what they do. Additionally, compounds for which a PubChem CID could not be identified were excluded from the analysis. For these compounds, the swissADME database cannot be used because the canonical SMILES is unknown.

Taken together, EESP suppressed DNFB-induced skin symptoms, histopathologic abnormalities and pro-inflammatory cytokines and chemokine without causing spleen shrinkage (evidence of systemic immunosuppression), weight gain suppression, or skin atrophy, as observed in the DEX-treated group. These results suggest the potential for the relatively safe dermatological application of *S. polyrhiza*.

## 4. Materials and Methods

### 4.1. Network-Based Pharmacological Analysis of S. polyrhiza

#### 4.1.1. Screening of Active Compounds and Related Targets in *S. polyrhiza*

Active compounds and related targets of *S. polyrhiza* were identified using the TCMBank (https://tcmbank.cn/ (accessed on 1 August 2023)) [29], TCMSP (Traditional Chinese Medicine Systems Pharmacology Database and Analysis Platform, https://old.tcmsp-e.com/tcmsp.php/ (accessed on 7 June 2023)) and ETCM (The Encyclopedia of Traditional Chinese Medicine, http://www.tcmip.cn/ETCM2/front/#/ (accessed on 14 August 2023)) database. Target names were checked using UniProt (http://www.uniprot.org/ (accessed on 8 June 2023)) [30] and analysed using CD-related gene information obtained from the DisGeNET platform (https://www.disgenet.org/ (accessed on 8 June 2023)) [31]. Absorption, distribution, metabolism, and excretion (ADME) parameters such as gastrointestinal (GI) absorption and drug-like nature for active compounds were predicted using the SwissADME database (http://www.swissadme.ch/ (accessed on 3 August 2023)). Inclusion criteria were high GI absorption and drug likeness greater than or equal to 0.55 [32].

#### 4.1.2. Network Analysis

Relationships between disease-related genes were analysed using STRING platform (https://string-db.org/ (accessed on 16 August 2023)). The network edge was displayed as a confidence level, and the minimum required interaction score was set to 0.7 (high confidence) or higher. The cytoHubba plugin was used for the identification of hub genes from the PPI analysis [33]. Cytoscape 3.9.1 (https://cytoscape.org/ (accessed on 16 August 2023)) was used to confirm compound-target networks [34]. For the identification of enriched transcriptomic signatures, an analysis was carried out using Enrichr website (https://maayanlab.cloud/Enrichr/ (accessed on 18 August 2023)).

#### 4.1.3. Docking Analysis

Docking analysis was performed as described previously [28], and the binding affinities of active compounds and target proteins were determined using AutoDock Vina [35]. The structures of the two proteins (PDB file) were obtained from the protein data bank (PDB, https://www.rcsb.org/ (accessed on 15 June 2023)) database, and the structures of the 14 active compounds were obtained from the PubChem (https://pubchem.ncbi.nlm.nih.gov/ (accessed on 15 June 2023)) database. Structures obtained for docking analysis were edited and visualized using AutoDockTools (ver 1.5.7, Molecular Graphics Laboratory, Greensboro, NC, USA) and PyMOL (The PyMOL Molecular Graphics System ver. 2.5.5, Schrödinger, New York, NY, USA) programs. The analysis process is summarized in Appendix A.

### 4.2. The Identification of Apigenin and Luteolin

#### 4.2.1. Preparation of EESP and the Standard Mixture

EESP was dissolved in methanol at 10 mg/mL. Apigenin, apigenin-7-glucoside, luteolin, and luteolin-7-glucoside were dissolved in methanol at 0.5 mg/mL. EESP and the standard mixture were filtered through a 0.20 μm Syringe Filter (Bio FACT TM, Daejeon, Republic of Korea) before HPLC.

#### 4.2.2. Chromatographic Conditions

HPLC was performed using an Agilent series 1100 instrument (Agilent, Waldbronn, Germany) equipped with a quaternary pump, a variable wavelength detector, and an autosampler. Samples were separated on a Capcell Pak C18 column (5 μm, 4.6 × 250 mm, Shiseido, Tokyo, Japan). The mobile phase consisted of mixtures of water containing 0.5% (*v*/*v*) formic acid (A) or 100% methanol (B). The following gradient elution program was used: 30–40% B for 0–20 min, 40–53% B for 20–30 min, 53–90% B for 30–40 min, and 90% B for 40–45 min. The unit was operated at a flow rate of 1.0 mL/min and a column temperature of 35 °C, and the UV detector was set at 254 nm. Data were processed using Agilent LC B.02.01 ChemStation software.

### 4.3. The Anti-Inflammatory Effects of EESP in Mice with CD

#### 4.3.1. Preparation of *S. polyrhiza* Extract

*S. polyrhiza* was purchased from the Kwangmyungdang Company (Ulsan, Republic of Korea), and a specimen was deposited in the School of Korean Medicine, Pusan National University (Voucher no. MS2017-006). *S. polyrhiza* was authenticated by Professor Hyungwoo Kim. In brief, whole *S. polyrhiza* (100 g) was extracted, as we previously described [26], and yielded 12.5 g of lyophilized powder (EESP, yield 12.5%). A sample of EESP was deposited in the School of Korean Medicine, Pusan National University (Voucher no. MH2017-006). 

#### 4.3.2. Animals

Male Balb/c mice (6 weeks old) were purchased from Samtaco (Osan, Republic of Korea). The mice were housed in a specific pathogen-free environment under a 12 h light/dark cycle and provided standard rodent food and water ad libitum. All animal experiments were conducted in accordance with the guidelines issued by the animal care and use committee of Pusan National University (approval number PNU-2019-2269).

#### 4.3.3. CD Induction and Experimental Schedule

CD was induced using DNFB, as we previously described [26]. In brief, mice were randomly divided into six groups, namely, a normal (NOR) group of non-treated mice (n = 6), a control (CTL) group of non-treated CD mice (n = 8), three EESP groups of CD mice treated with 50, 150, or 500 μg/day of EESP for six consecutive days (n = 8/group), and a DEX-treated CD group treated with 150 μg/day of DEX for six consecutive days (n = 8). For CD induction, mice were sensitized by the application of 30 µL DNFB (0.2%, *v*/*v*) in acetone: olive oil (AOO, 4:1) onto both ears for three consecutive days. Five days after sensitization, each mouse was challenged by the application of 60 µL DNFB (0.2%, *v*/*v*) in AOO onto the shaved back every two days (4 times). EESP and DEX were dissolved in 70% ethanol, diluted in AOO, and then applied onto the shaved backs of mice. The experimental schedule is summarized in the Appendix A.

#### 4.3.4. Skin Surface Scores and Thicknesses Measurements

Skin lesions were imaged using a digital camera (Olympus, Tokyo, Japan), and skin surface scores were evaluated using the modified version of Amano’s method [36]. In brief, the skin surface score was determined by summing the scores for scaling, excoriation, erythema, and skin roughness assessed using a 4-point scale (0, none; 1, mild; 2, moderate; 3, severe). Skin tissues were cut into 5 mm diameter pieces, and weights were measured using an electronic scale (Sartorius, Seongnam, Republic of Korea).

#### 4.3.5. Erythema and Melanin Indices

Erythema and melanin indices were determined using a dermo-spectrophotometer (Cortex Technology, Hadsund, Denmark) at three different skin locations per mouse.

#### 4.3.6. Histopathological Examinations 

Skin tissues were fixed in 10% (*v*/*v*) formaldehyde, embedded in paraffin, sectioned at 4 μm, and stained with hematoxylin and eosin, and histological changes were observed under a light microscope (Carl Zeiss AG, Oberkochen, Germany) at ×50.

#### 4.3.7. Evaluations of Epidermal Hyperplasia and Immune Cell Infiltration

Vertical distances between the basal lamina and outer stratum granulosum were measured to evaluate epidermal hyperplasia. Three random measurements were made per slide using the Zen program (ZEISS, Jena, Germany). Cells were enumerated using a cell-counting grid (2.1 × 1.6 mm, 100× magnification) to evaluate immune cell infiltration in four randomly selected, non-overlapping regions per slide. Immune cells were defined as macrophages, polymorphonuclear leukocytes, lymphocytes, eosinophils, plasma cells, and giant cells [37]. The number of infiltrated immune cells was represented as the number per square millimeter.

#### 4.3.8. Measurement of Inflammatory Cytokines and Chemokine

Skin tissues (30 mg) were lysed in 300 μL PRO-PREP protein solution (iNtRON, Seongnam, Republic of Korea) using a bullet blender (Next Advance, Averill Park, NY, USA). Levels of TNF-α, IFN-γ, IL-6, and MCP-1 in lysates (50 μg) were determined using the mouse inflammation cytometric bead array kit (B.D. Biosciences, San Diego, CA, USA). 

#### 4.3.9. Western Blot Analysis

Protein isolation and Western blotting were performed as previously described [28]. Briefly, each primary antibody was diluted in tris-buffered saline with 0.1% Tween^®^ 20 Detergent (TBST) containing 5% skim milk at a ratio of 1:1000, and reacted overnight at 4 °C. The list of primary antibodies used is as follows: P-ERK (#sc-7383, Santa Cruz Biotechnology, Dallas, TX, USA), ERK2 (#sc-1647, Santa Cruz Biotechnology, Dallas, TX, USA), p-p38 (#4631, Cell Signaling Technology, Danvers, MA, USA), P38 (#9212, Cell Signaling Technology, Danvers, MA, USA), p-JNK (#4671, Cell Signaling Technology, Danvers, MA, USA), JNK (#9258, Cell Signaling Technology, Danvers, MA, USA) and β-actin (#sc-47778, Santa Cruz Biotechnology, Dallas, TX, USA). Secondary antibody (Goat anti-mouse IgG F(ab′)2, polyclonal antibody (HRP conjugate), Enzo Life Sciences, Farmingdale, NY, USA) was diluted in TBST containing 5% skim milk at a ratio of 1:2000, and reacted for 2 h at room temperature. Blots were visualized using an enhanced chemiluminescence system (ImageQuant™ LAS 4000, ver. 1.3, General Electric Company, Boston, MA, USA).

#### 4.3.10. Statistical Analysis

Data were analysed using One-way ANOVA followed by Dunnett’s multiple comparison test. Prime 5 for Windows version 5.01 (GraphPad Software Inc., La Jolla, CA, USA) was used for the analysis. Results are presented as means ± standard deviations (SDs), and statistical significance was accepted for *p* values < 0.05. 

## 5. Conclusions

Network-based analysis suggested 14 active compounds ((+)-armepavine, (R)-N-methylcoclaurine, 2-hexadecenoic acid, anonaine, apigenin, butylated hydroxytoluene, D-ascorbic acid, higenamine, isoliensinine, luteolin, nornuciferine, nuciferine, palmitic Acid and roemerine) and 29 CD-related genes (AHR, BCL2, CASP8, CCL2, CCN2, CYP1A1, DDIT3, EPHB2, GCLC, GSR, GSTP1, HMOX1, IFNG, IL1A, IL1B, IL4, IL6, MAPK1, MMP10, OPRM1, S100A8, SOD1, SOD3, TLR4, TNF, TPSAB1, UGT1A1, UGT1A6, VEGFA) in *S. polyrhiza*. Among them, TNF and IL6 were identified as hub genes. Among 14 active compounds, luteolin and apigenin showed strong binding affinities with both TNF and IL-6. In addition, EESP contained a relatively high amount of luteolin compared to apigenin. Furthermore, our in vivo study showed that EESP inhibited the ERK pathway and the production of TNF-α, IFN-ɣ, IL-6, and MCP-1 in inflamed tissues, suppressed DNFB-induced histopathological abnormalities and immune cell infiltration, and reduced the severity of skin lesions. In summary, our findings suggest the potential for dermatological applications of *S. polyrhiza,* and that its anti-dermatitis activity is related to the inhibition of TNF and IL-6 by luteolin and luteolin glycosides.

## Figures and Tables

**Figure 1 ijms-24-13271-f001:**
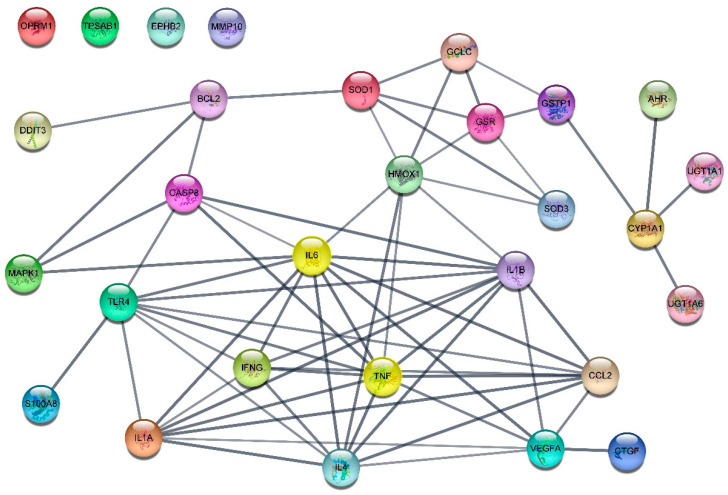
Relationship between 29 CD-related targets of *S. polyrhiza.* The edges indicate both functional and physical protein associations and line thickness indicates the strength of data support.

**Figure 2 ijms-24-13271-f002:**
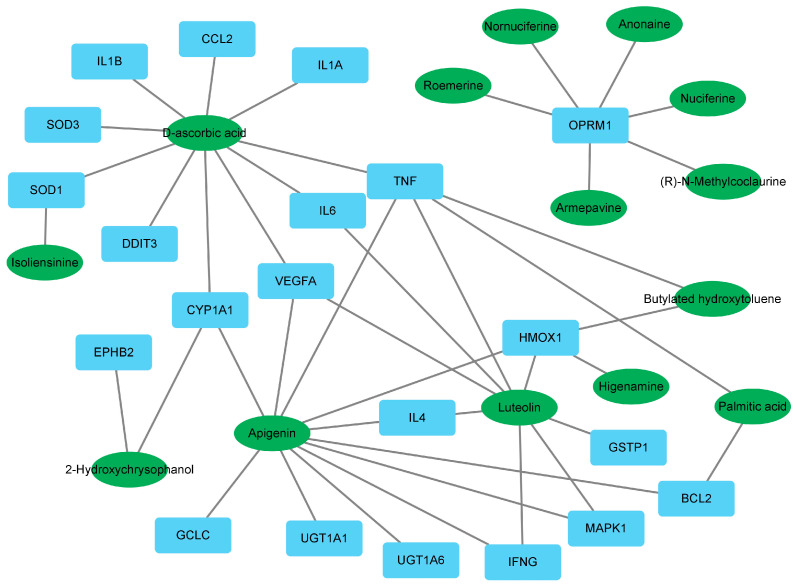
Correlations between 14 active compounds and the 21 CD-related targets. Green indicates the active compounds, and light blue indicates the target genes.

**Figure 3 ijms-24-13271-f003:**
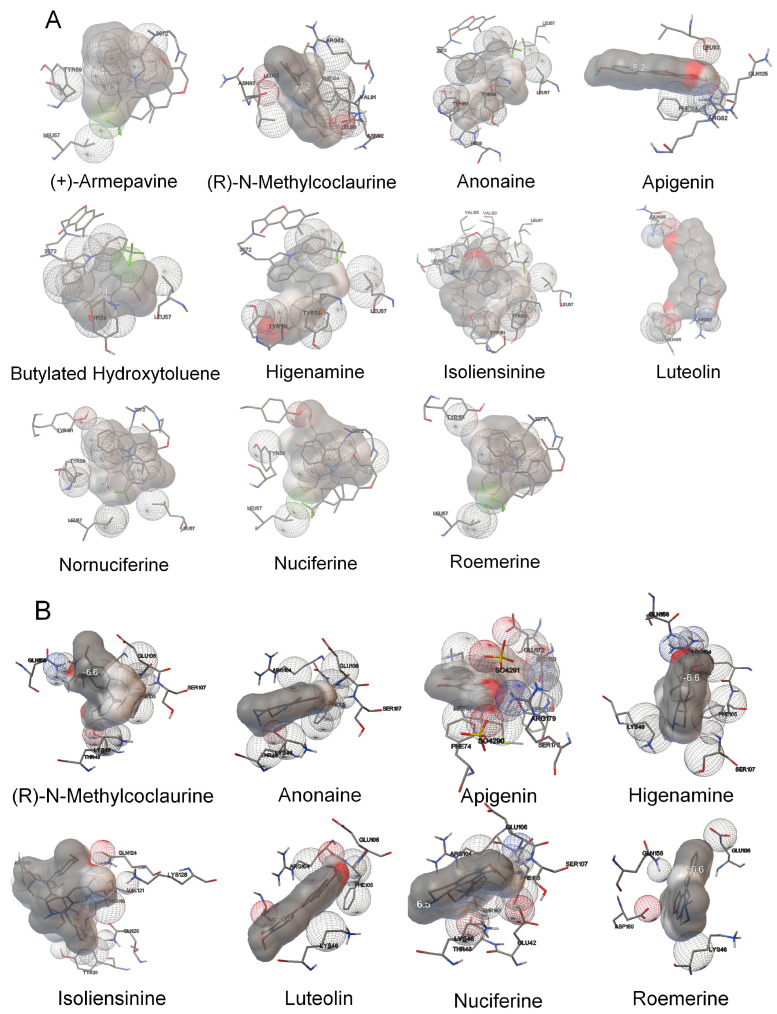
Binding modes of the active compounds and residues of two proteins. Eleven active compounds and residues of TNF (**A**) and eight active compounds and residues of IL-6 (**B**). Gray cartoons indicate active compounds.

**Figure 4 ijms-24-13271-f004:**
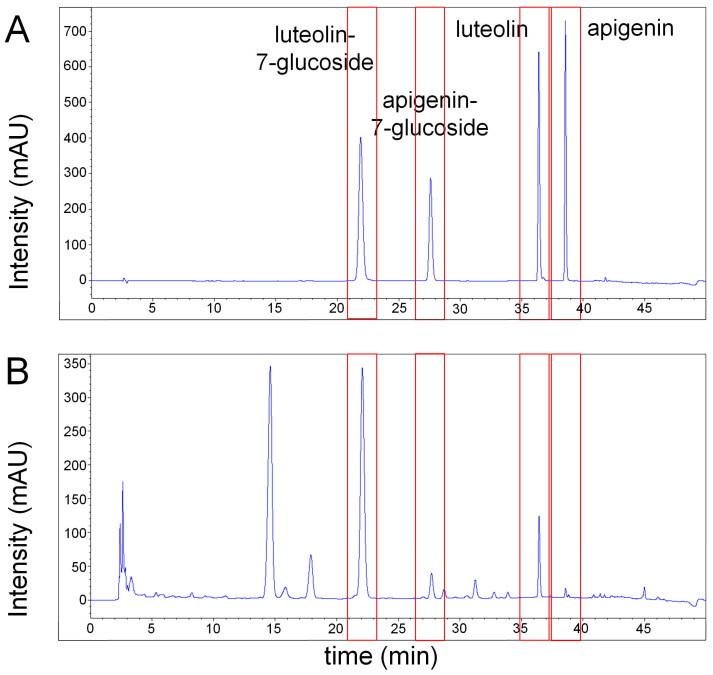
HPLC chromatograms of a standard mixture containing apigenin, apigenin-7-glucoside, luteolin, and luteolin-7-glucoside and EESP. The chromatograms of apigenin, apigenin-7-glucoside, luteolin, and luteolin-7-glucoside in a standard mixture (**A**) and in EESP (**B**) were detected at a UV wavelength of 254 nm.

**Figure 5 ijms-24-13271-f005:**
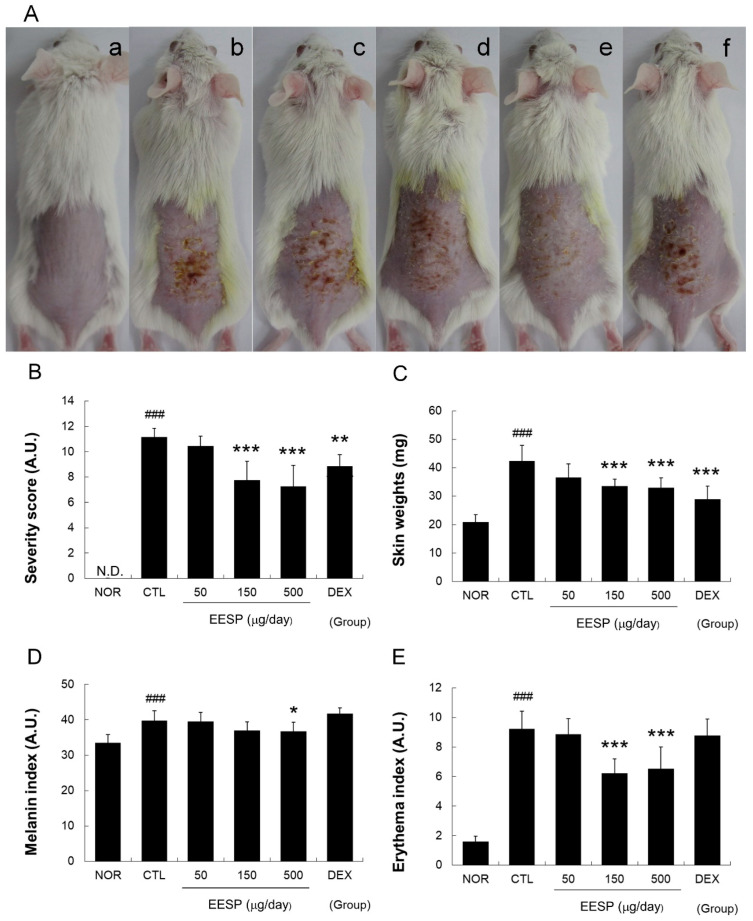
Effects of EESP on skin lesions, weight, and color in CD mice. Skin lesions were imaged using a digital camera. (**a**), Treatment-naïve (NOR); (**b**), CD control (CTL); (**c**), 50 µg/day EESP; (**d**), 150 µg/day EESP; (**e**), 500 µg/day EEFR; and (**f**), 150 μg/day DEX (**A**). Severity scores were estimated semi-quantitatively (**B**). Weights of 5 mm diameter skin samples were measured using a micro-balance (**C**). Erythema and melanin indices were determined using a dermo-spectrophotometer (**D**,**E**). EESP, ethanolic extract of whole *Spirodela polyrhiza*; DEX, dexamethasone; N.D., undetectable. Results are presented as means ± standard deviations (SDs). ^###^
*p* < 0.001 vs. NOR; * *p* < 0.05, ** *p* < 0.01 and *** *p* < 0.001 vs. CTL, N.D. means undetectable.

**Figure 6 ijms-24-13271-f006:**
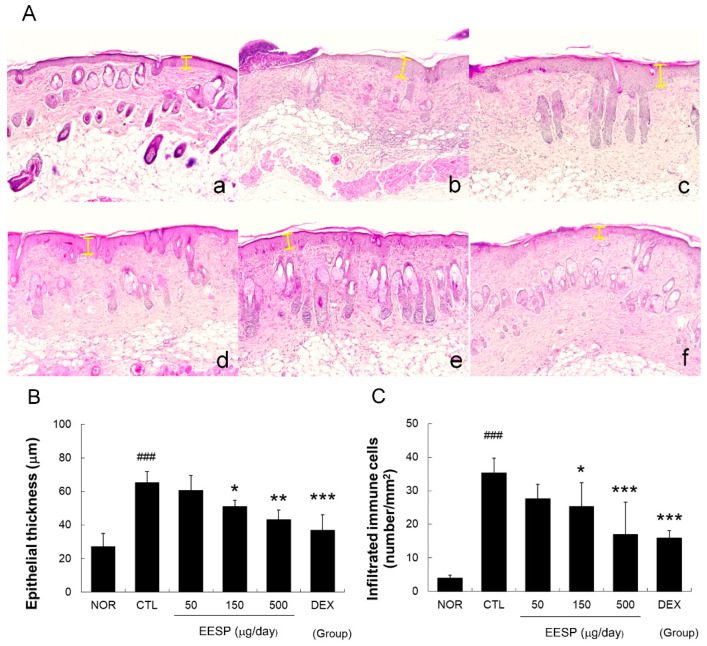
Effects of EESP on histopathological abnormalities in inflamed tissues. The presented sequence is the same as that in Figure 5A. The yellow bars indicate epidermal thicknesses (original magnification ×50) (**A**). Histogram showing mean epidermal thicknesses (**B**) and numbers of infiltrating immune cells (**C**) in the six study groups. Abbreviations are as defined in Figure 5. Results are presented as means ± SDs. ^###^
*p* < 0.001 vs. NOR; * *p* < 0.05, ** *p* < 0.01 and *** *p* < 0.001 vs. CTL.

**Figure 7 ijms-24-13271-f007:**
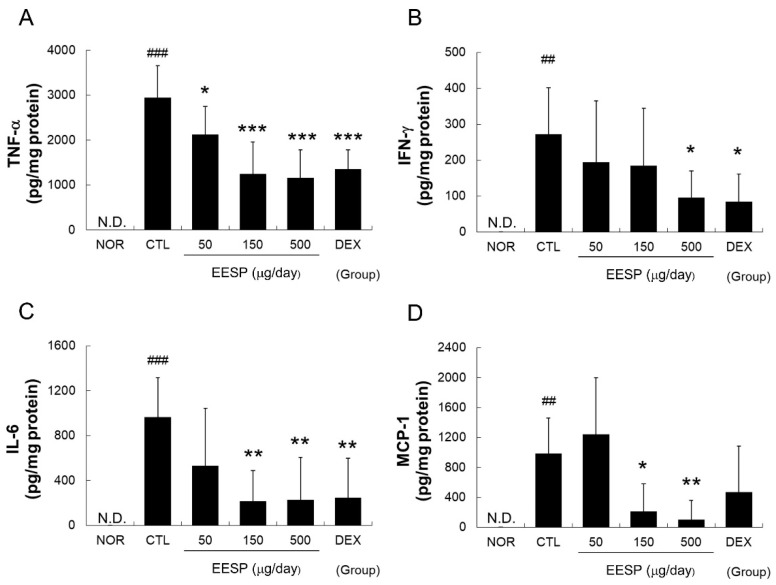
Effects of EESP on cytokine and chemokine levels in inflamed tissues. Levels of cytokines and chemokine were evaluated using a cytometric bead array. (**A**) TNF-α; (**B**) IFN-γ; (**C**) IL-6; (**D**) MCP-1. N.D. means undetectable. Abbreviations are as defined in Figure 5. Results are presented as means ± SDs. ^##^
*p* < 0.01 and ^###^
*p* < 0.001 vs. NOR; * *p* < 0.05, ** *p* < 0.01 and *** *p* < 0.001 vs. CTL.

**Figure 8 ijms-24-13271-f008:**
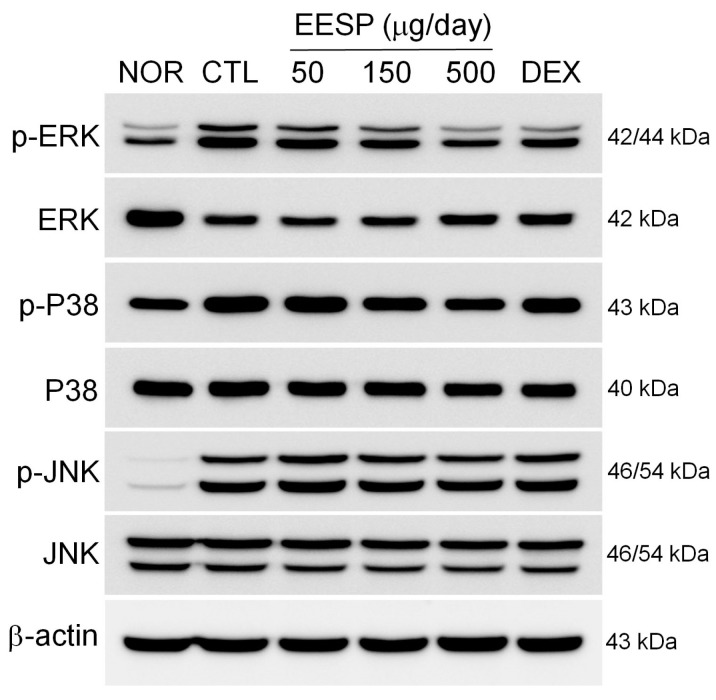
Effects of EESP on DNFB-induced phosphorylation of MAPK signalling pathways in inflamed tissues. MAPK signalling molecules were determined by Western blot. Abbreviations are as defined in Figure 5.

**Table 1 ijms-24-13271-t001:** Active compounds of *S. polyrhiza*.

No.	Molecule Name	PubChem CID	DrugBank Accession No.	Molecular Formula	Structure	Molecular Weight (g/mol)
1	(+)-Armepavine	680292	-	C_19_H_23_NO_3_	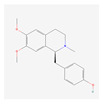	313.39
2	(R)-*N*-Methylcoclaurine	440595	-	C_18_H_21_NO_3_	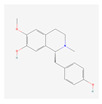	299.36
3	2-Hexadecenoic acid	5282743	-	C_16_H_30_O_2_	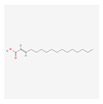	254.41
4	Anonaine	160597	-	C_17_H_15_NO_2_	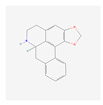	265.31
5	Apigenin	5280443	DB07352	C_15_H_10_O_5_	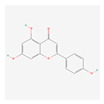	270.24
6	Butylated Hydroxytoluene	31404	DB16863	C_15_H_24_O	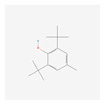	220.35
7	D-ascorbic acid	54690394	DB00126	C_6_H_8_O_6_	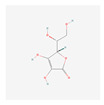	176.12
8	Higenamine	114840	DB12779	C_16_H_17_NO_3_	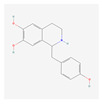	271.31
9	Isoliensinine	5274591	-	C_39_H_45_NO_6_	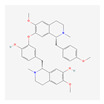	610.7
10	Luteolin	5280445	DB15584	C_15_H_10_O_6_	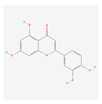	286.236
11	Nornuciferine	41169	-	C_18_H_19_NO_2_	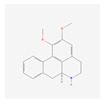	281.35
12	Nuciferine	10146	-	C_19_H_21_NO_2_	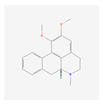	295.4
13	Palmitic Acid	985	DB03796	C_16_H_32_O_2_	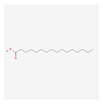	256.42
14	Roemerine	119204	-	C_18_H_17_NO_2_	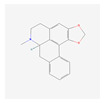	279.3

**Table 2 ijms-24-13271-t002:** Binding affinities for interactions between two proteins (TNF and IL-6) and 14 active compounds.

Molecule Name	Affinity (kcal/mol)	
	TNF (PDB ID, 2AZ5)	IL-6 (PDB ID, 1ALU)
Roemerine	−9.6	−6.6
Anonaine	−9.4	−7.0
Luteolin	−8.9	−6.4
Nornuciferine	−8.9	−5.9
Nuciferine	−8.7	−6.5
Apigenin	−8.2	−6.6
Higenamine	−8.0	−6.6
Isoliensinine	−8.0	−6.4
(+)-Armepavine	−7.7	−5.7
(R)-N-Methylcoclaurine	−7.2	−6.6
Butylated Hydroxytoluene	−7.1	−5.4
Palmitic Acid	−5.6	−3.7
2-Hexadecenoic acid	−5.4	−3.7
D-ascorbic acid	−5.2	−5.4

PDB means protein data bank.

## Data Availability

The data presented in this study are available upon request from the corresponding author.

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
