# Peer review of "Anti-Inflammatory Effects of Spirodela polyrhiza (L.) SCHLEID. Extract on Contact Dermatitis in Mice—Its Active Compounds and Molecular Targets"

_ijms, 2023, doi:10.3390/ijms241713271_

Round 1

Reviewer 1 Report

First of all, I would like to thank the editor and the journal for inviting me to review the manuscript for this prestigious journal.

As a result of reviewing the requested manuscript (ijms-2501528) for a considerable amount of time, I have reached the following opinion.

#Recommendations

#Major Concerns

1. A problem with multiple network pharmacology studies is that reliable or consistent predictions may not be made depending on the source of information used. Methodological considerations are needed to improve this. Here are some things that can improve this manuscript from this point of view.

1.1. The condition of the database for the collection of potentially active compounds is "basically it should be a well updated platform". TCMSP is a good database that provides a lot of convenience for screening in that it systematically provides ADME information, but it is no longer possible to guarantee the accuracy of predictive analysis using only this database in terms of the quality and quantity of information received and updated. In fact, when searching for the compound contained in the material of this manuscript in ETCM, another excellent and widely used database, (7Z,10Z,13Z-Hexadecatrienoic Acid, Hexadecatrienoic Acid, Apiose, 11Z-Hexadecenoic Acid, (10R)-Hydroxyhexadeca-7Z, 11E,13Z-Trienoic Acid, (4R)-4-Hydroxyisophytol, 8-Hydroxyluteolin-8-Î'-D-Glucopranoside, Isoscoparin, Trans-1,3-Phytodiene) etc. can be tested. Therefore, it is considered better to achieve the goal of network pharmacology analysis by re-collecting compounds using at least two other databases besides TCMSP and performing the analysis from there.

Please review the following recommended up-to-date references to help you in your work.

suggested ref 1. Li X, Liu Z, Liao J, Chen Q, Lu X, Fan X. Network pharmacology approaches for research of Traditional Chinese Medicines. Chin J Nat Med. 2023 May;21(5):323-332. doi: 10.1016/S1875-5364(23)60429-7

suggested ref 2. Zhao L, Zhang H, Li N, Chen J, Xu H, Wang Y, Liang Q. Network pharmacology, a promising approach to reveal the pharmacology mechanism of Chinese medicine formula. J Ethnopharmacol. 2023 Jun 12;309:116306. doi: 10.1016/j.jep.2023.116306

1.2. For ADME-based screening of newly collected compounds, it is recommended to consider both the results of simulations using SwissADME (http://www.swissadme.ch/) and the selection criteria provided by TCMSP. The following references can help you with this task.

Suggested Ref. Daina A, Michielin O, Zoete V. SwissADME: a free web tool to evaluate pharmacokinetics, drug-likeness and medicinal chemistry friendliness of small molecules. Sci Rep. 2017;7:42717. doi:10.1038/srep42717

1.3. Use Uniprot for cross-validation and standardization of already collected gene targets. It is recommended to run prediction simulations separately for each credible gene targets of candidate compounds. The following references may be helpful.

Suggested Ref. Yao ZJ, Dong J, Che YJ, et al. TargetNet: a web service for predicting potential drug-target interaction profiles via multi-target SAR models. J Comput Aided Mol Des. 2016;30(5):413-424. doi:10.1007/s10822-016-9915-2

1.5. The DrugBank database (https://go.drugbank.com/, version 5.0) is the most reliable DB containing human genome data on which drugs already approved by the FDA. Therefore, this DB must be included when searching for disease-related targets. Please use the following references.

Suggested Ref. Wishart DS, Feunang YD, Guo AC, et al. DrugBank 5.0: a major update to the DrugBank database for 2018. Nucleic Acids Res. 2018;46(D1):D1074-D1082. doi:10.1093/nar/gkx1037

1.6. Doesn't network pharmacology usually use the intersection of the entire genomic target pool of druggable compounds contained in the target natural product and the entire disease-related genomic target for analysis? However, in this study, after screening 20 compounds, only 5 compounds were selected whose gene targets were found in the DisGeNET database. From a network science point of view, this method may introduce a bias that may affect the accuracy of the analysis or may run the risk of not revealing all the potential mechanisms of the herb being analyzed. What previous research has informed this methodology?

1.7. When reporting the results of a PPI network analysis such as Figure 1, please report the number of nodes and edges for each node, the metrics for each node (such as degree centrality), and the screening criteria for important hub genes. On the other hand, in the case of GSTP1, which has no association with other gene targets in the analysis of Figure 1, please provide the reason for exclusion in the text before excluding the analysis.

1.8. In this manuscript, you did not perform GO and KEGG enrichment analysis for the hub gene target of the studied herb? why didn't you? It is considered appropriate to add this analysis to obtain more predictive information and to identify important pathways.

1.9. Why was molecular docking performed on only one target (TNF)? Line 95 states that at least TNF and IL-6 are hub gene targets, but there is no adequate explanation for this inconsistency. Meanwhile, IL-6 is used as a target for experimental research later in this manuscript. And separately, the results of the analysis of the hydrogen bonds with amino acid residues and the distance of each target compound must be reported without omission.

2. The HPLC results finally identified apigenin and luteolin. Is the pharmacological activity of the herb in this study mainly dependent on these two compounds? Is there room for these two types of compounds to be used as efficacy-based indicators? Authors should provide detailed answers to these two questions to the readers in the Discussion section.

3. Looking at Figure 5E, EESP showed a dose-dependent result, while dexamethasone, an active control group with strong anti-inflammatory effect, did not show any results. Even if it is possible for the melanin index, it is difficult to understand that the DEX group did not show a significant effect on the erythema index. Please let us know what the author thinks about this.

4. All types of biomedical research, whether non-clinical or clinical, must have a clear target "unmet medical need". As far as I know, contact dermatitis is not necessarily treated with glucocorticoids alone. So what are the most pressing problems in contact dermatitis that can be addressed by the herb in this trial and that are currently unmet? The context related to this question should be clearly presented in the introduction, and a clear answer to this question should be presented in one sentence in the conclusion.

5. (Line 284-285) "These findings imply that the anti-inflammatory mechanism of EESP differs from that of corticosteroids, especially in terms of systemic immune suppression."

What could support this sentence should be explained to the reader more specifically in the discussion section. From the same point of view, what better benefits EESPs can provide to patients with contact dermatitis than glucocorticoids should be presented more clearly. Of course, the author's insights on additional research for this should be suggested.

6. (Line 257) "MAPK signaling plays an important role in the pathogenesis and progression of CD (Figures 1 and 2) and in cytokine secretion."

The information presented in this study alone does not support the claim that EESP exerts its therapeutic effect primarily through the MAPK pathway.

7. At this time, the proposition of L388-390 is not supported by the data presented in this study. After sufficient revision of the entire manuscript, the conclusion will need to be re-examined.

# Minor issues

1. The significance of network pharmacology methodology does not seem to need to be introduced in the introduction of this manuscript. Network pharmacology is already widespread and is a generalized technique, especially in natural product research. Therefore, it is sufficient to present only the combination of pharmacological prediction and experimental verification in 2-3 sentences to examine the specificity of the material itself and any promising pharmacological effects of the material in this manuscript.

2 Discussion The first paragraph should summarize and briefly present the main findings of the entire study and their significance. It is sufficient to briefly mention in the introduction the traditional experience of using the herb under study. At the same time, there is no need to devote a paragraph to a discussion of the general advantages of network pharmacology. Unless there is a very unusual methodological advance, the discussion section should simply provide a logical basis for the observations of this manuscript.

3. The entire discussion is unreadable and confusing. Please reorganize the entire content according to the following table of contents.

1) Summary and significance of the results of this study

2) Whether the pharmacological effect of EESP, which can be generally confirmed in network pharmacology and experimental studies, is supported by previous studies.

3) Detailed analysis by citing previous studies on the significance of the effect and effect size of the indicators shown in this experimental study.

4) methodological strengths and limitations of the study

5) applicability of this study and suggestions for follow-up studies

4. Normally, the ADME selection criteria with the TCMSP DB are OB 30% and DL 0.18, which are used in many studies. Why was 20%/0.1 used as the selection criterion in this study?

5. The content of the Methods section related to Network Pharmacology Analysis is too concise and does not contribute to the reproducibility of the study. Substantial revision is needed. Please refer to the well-documented references below.

Suggested Ref. Yang N, Shao H, Deng J, Liu Y. Network pharmacology-based analysis to explore the therapeutic mechanism of Cortex Dictamni on atopic dermatitis. J Ethnopharmacol. 2023 Mar 25;304:116023. doi: 10.1016/j.jep.2022.116023

It is my hope that my opinion will contribute to the successful publication of this manuscript.

It is necessary to proofread the entire manuscript by an expert who has both professional academic publishing experience and native English proficiency.

Author Response

All issues raised have been thoroughly reviewed, and the manuscript has been revised based on comments.

Reviewer 2 Report

English grammar problems require minor revision.

Author Response

(The authors gave the same response as above.)

Round 2

Reviewer 1 Report

The author has been diligent in answering my questions, and the manuscript has improved considerably.

However, there are still some aspects that are questionable, such as 

# Major concerns

1) The reliability of the list of active compounds in HERB can be overcome by using some additional good quality databases. (Of course, ETCM for example has recently undergone a major update to ETCM 2.0 and should have been used). However, I have said in previous reviews that the reliability and comprehensiveness of each compound is not solved by an established DB alone. 

For example, looking at Supplementary File 2, I can see that No. 2 "(E,7S,11R)-3,7,11,15-tetramethylhexadec-2-en-1-ol" corresponds to PubChem CID 6474938. (Providing PubChem CID numbers when presenting a compound list is very important for data completeness. ) The target prediction for this molecule predicts potential activity against several classes of targets such as Nuclear receptor, Phosphatase, Ligand-gated ion channel, Kinase, including Androgen Receptor (AR), Glycine receptor subunit alpha-1 (GLRA1), Dual specificity phosphatase Cdc25B (CDC25B), Protein kinase C gamma (PRKCG). However, file 2 above states that there is no target with PubChem CID 6474938. 

How can the reliability and reproducibility of the targets of potential compounds in this study be ensured if these problems are still found? 

2) (Line 112-113) Among them, TNF and IL6 play important roles in the induction and exacerbation of CD (Figure 1 and Supplementary Data 4).

It is very questionable whether the above sentence can be expressed in such a strong tone. Looking at Supplementary Data 4, the raw data are only about the interaction and the number of edges. At this point, the following questions need to be answered. 

2-1) Basically, the criteria for picking out gene targets that are at least "more deserving" of attention in PPI is social network analysis, which is why terms like interaction and edge are used. And the main quantitative criterion for judging this is centrality. How did you calculate the concept of centrality to determine the criterion of "importance"? Is betweenness centrality important in this study? Is closeness centrality important? Or is prestige centrality important? Have you considered using MCC or MNC? Why did you choose this centrality measure? These are various questions about PPI analysis that readers may have. The method and results of the manuscript should be strengthened so that these questions never arise. 

2-2) A high centrality concept does not necessarily mean that the target is biologically important, so the tone is more scientifically appropriate to say "this is a target that deserves priority consideration because of its high centrality", not necessarily the same as "important".

To address the above issues, I would recommend that the authors refer to at least five other network pharmacology studies.

3) What is in Supplementary Data 6 deserves a method and a graph. At the very least, it should be mentioned in passing in the discussion without any mention of the method, even though some analysis was done. The authors state that the number of targets used in this study was too small to do an enrichment analysis, which I don't quite understand, and I would like to see a rationale for this judgment. 

In addition, if the KEGG enrichment results confirm that, for example, the MAPK pathway is important, one should perform a KEGG mapping analysis using a list of A) drug-disease overlapping gene targets, B) drug-disease non overlapping gene targets, and C) disease non overlapping gene targets, respectively, and compare these results with the experimental results in the discussion. Similarly, the results of the enrichment analysis as a whole, compared with the experiments, are also important information to cover in the discussion, with similarities as strengths and differences as limitations.

# minor issues 

1) When you open Supplementary file 2, it is labeled as Supplementary data 5. Please check that this confusion does not exist in any other files and correct it. 

2) The authors simply state (line 61-62) "In particular, it is more necessary when repeated use is unavoidable, such as in the case of occupational skin diseases". Do you really think that this sentence is an optimal statement of the most important unmet medical need for the disease addressed in this study? Is this really a reason to convince a decision maker or health care provider to fund the trial? Findings from epidemiological studies or relevant clinical trials should be cited. If there is a lack of information about the actual human problems associated with the target disease, how can you explain to the reader that the results of the experiment will benefit humans in any way? 

3) The introduction section is still confusing overall, especially since it is inappropriate to spend 2 paragraphs on relevant DBs alone unless this manuscript is about remarkable advances in the methodology of network pharamcolgy. It would be sufficient to transfer the characteristics of the network pharmacology-related DBs to the Methods, and any idiosyncrasies can be explained in the Discussion as support for the methodological excellence of this manuscript. 

In the meantime, please utilize the references below regarding how to write a concise yet scientific introduction section. 

suggested ref. Cals JW, Kotz D. Effective writing and publishing scientific papers, part III: introduction. J Clin Epidemiol. 2013 Jul;66(7):702. doi: 10.1016/j.jclinepi.2013.01.004

4) The Discussion section is also too many paragraphs and does not have a good logical flow. Considering the data in this manuscript, it should be condensed to a maximum of 4-5 paragraphs, and the length should be reduced to about 3/4 of the current length. The section needs to be improved so that it coherently supports the conclusions.

It would be good to utilize the following references 

Suggested ref. Ghasemi A, Bahadoran Z, Mirmiran P, Hosseinpanah F, Shiva N, Zadeh-Vakili A. The Principles of Biomedical Scientific Writing: A Discussion. Int J Endocrinol Metab. 2019 Jul 29;17(3):e95415. doi: 10.5812/ijem.95415

I hope that my comments will contribute to the successful publication of the manuscript, and I am willing to continue to invest my personal time to discuss it.  

Since this manuscript still requires considerable discussion before publication, it would be pointless to comment on the quality of the English proofreading at this point.

Author Response

Thank you for your thorough review of this manuscript. The changes made in the second round are highlighted in blue.

Reviewer 2 Report

can be acceptted

ok

Author Response

(The authors gave the same response as above.)

Round 3

Reviewer 1 Report

Through the hard work of the authors, this manuscript has been greatly improved, and at the same time, all questions have been answered. 

I consent to the publication of this manuscript in its current state.